# The experience of a program combining two complementary therapies for women with breast cancer: An IPSE qualitative study

Jordan Sibeoni[1,2,3]*, Emilie Manolios[2,3,4], Jeanne Mathé[2,3], Valérie Feka[5], Marie-Madeleine Vinez[3], Evelyne Lonsdorfer-Wolf[5], Jean-Gérard Bloch[6,7], Franck Baylé[2,8], Jean-Pierre Meunier[3], Anne Revah-Levy[1,2,3], Laurence Verneuil[2,3,8]

1 Pôle Psychiatrie et Santé Mentale, Centre Hospitalier Victor Dupouy, Argenteuil, France, 2 INSERM U1153, Statistic and Epidemiologic Research Center Sorbonne Paris Cité, (CRESS), ECSTRRA Team, Université Paris Cité, Paris, France, 3 IPSEA: IPSE Association, IPSEA.fr, Paris, France, 4 AP-HP, Service de Psychiatrie et Addictologie de l'adulte et du Sujet âgé, Hôpital Européen Georges-Pompidou, Paris, France, 5 Service de Physiologie et EFR, NHC, Hôpitaux Universitaire de Strasbourg, Strasbourg, France, 6 Service de Rhumatologie, Hôpitaux Universitaires de Strasbourg, Strasbourg, France, 7 Institut Français Pleine Conscience Mindfulness Strasbourg, Strasbourg, France, 8 GHU Paris Psychiatrie-Neurosciences, Pole Précarité, Hopital Sainte Anne, Paris, France

* jordansib@hotmail.com

**Data Availability Statement:** Data cannot be shared publicly because of ethical restrictions. Data are available from the IPSE Association, at

## Abstract

### Introduction

The use of complementary therapies within oncology is a clinical issue, and their evaluation a methodological challenge. This paper reports the findings of a qualitative study exploring the lived experience of a French program of complementary therapies combining structured physical activity and MBSR among women with breast cancer.

### Methods

This French exploratory qualitative study followed the five stages of the Inductive Process to analyze the Structure of lived Experience (IPSE) approach. Data was collected from February to April 2021 through semi structured interviews. Participants, purposively selected until data saturation. Inclusion criteria were: being an adult woman with breast cancer whatever the stage who had completed their treatment and were part of the program of complementary therapies.

### Results

29 participants were included. Data analysis produced a structure of experience based on two central axes: 1) the experience these women hoped for, with two principal expectations, that is to take care of their bodies and themselves, and to become actors in their own care; and 2) an experience of discovery, first of themselves and also in their relationship with the exterior, whether with others, or in society, and in the relationships with health-care providers.

ipseassociation@gmail.com, for researchers who meet the criteria for access to confidential data.

**Funding:** The authors received no specific funding for this work.

**Competing interests:** The authors have declared that no competing interests exist.

**Abbreviations:** CT, complementary therapies; IPSE, Inductive Process to analyze the Structure of lived Experience; MBSR, mindfulness-based stress reduction; PPIs, positive psychology interventions; QoL, quality of life.

## Conclusions

Our results from this French study reinforce the data described in other western countries about the needs of women receiving care in oncology departments for breast cancer: they need to be informed of the existence of supportive care in cancer by the health-care professionals themselves, to be listened to, and to receive support care. A systematic work of reflexivity about this redundancy in our results and in the qualitative literature, led us to question what impeded the exploration of more complex aspects of the experience of this women —the inherently emotional and anxiety-inducing experience of cancer, especially anxiety about its recurrence and of death–and to suggest new research perspectives to overcome these methodological and theoretical obstacles.

## Introduction

Breast cancer is the most frequently diagnosed cancer in women worldwide (nearly 2.2 million cases in 2020) (*WHO, March 2021*, [1]) as well as the leading cause of death among women. Survival has improved significantly due to medical advances (in diagnosis, surgery, radiation therapy, and new treatments, such as molecular targeted agents). Despite this significant progress, a large proportion of patients with cancer experience a deterioration of their quality of life, linked to effects of the disease and/or side effects of its treatments [2]. Supportive care in cancer, defined as "all of the care and support needed by patients during and after the disease, associated with specific treatments against cancer when these are implemented [3]," is based on global support of patients and qualifies all the care that enables them to manage the consequences of both the disease and its treatments. It improves patients' quality of life [4, 5] and is now an integral part of modern oncology care [6]. Recourse to complementary therapies is extremely frequent in supportive cancer care. These therapies constitute a set of knowledge, practices, and skills based on indigenous experiences, beliefs, and theories from different cultures—whether or not they can be explained—and are used both for health maintenance and for the prevention, diagnosis, improvement or treatment of physical or mental illnesses [7]. For women with breast cancer, the quantitative literature on supportive care shows symptomatic efficacy—against fatigue, anxiety and depression, sleep disorders, attention and memory, as well as pain–and improved quality of life (QoL) associated with various complementary therapies [8, 9], including but not limited to sports activity [10], mindfulness-based stress reduction (MBSR) [8, 11], and yoga [12]. This comprehensive multidisciplinary approach remains necessary and useful for several years after diagnosis [13]. Clinical practice guidelines from the Society for Integrative Oncology on the use of integrative therapies during and after breast cancer treatment recommend especially mind-body therapies but do not give any clinical indications or factors (age, stage of cancer, socioeconomic status) to choose among the many supportive care strategies [14]. However, one qualitative study conducted in United States has shown the influence of socio-ecological and cultural factors (beliefs about the illness, gender roles and family obligations) on the health-related quality of life of women with breast cancer [15].

In France, access to supportive care in cancer is less developed than in other places in the world [16]. According to a survey conducted by the French national institute of cancer (INCA), only 23% of the patients were offered such supportive care, most of the time at an advanced stage of their disease [17]. Another national survey has found an important

heterogeneity in the organization and accessibility of such care in France and a lack of assessment of the patients' needs [18]. Similarly, traditional complementary and integrative medicine use by patients with cancer in Western Europe is estimated to be 37% with important variations across countries, with a higher prevalence in German-speaking countries [19]. At the same time, some of these complementary therapies, beyond the field of supportive care in cancer, are quite established in France (e.g homeopathy) or in full expansion on the whole national territory (e.g MBSR).

In 2015, women with breast cancer being treated in oncology departments of the university hospital of Strasbourg, situated at the border with Germany, were invited to join a supportive care program combining two complementary therapies: MBSR and physical activity within the context of a prospective interventional study. The program comprised 2 sessions/week of a tailored endurance training exercise on a stationary bicycle, supervised and monitored by the same nurse for 8 weeks, after or simultaneously with a standard MBSR program (8 sessions, once a week, of group meditation combined with assigned individual exercises with 40 minutes of meditation/day. The participants began the program after having completed surgery, chemotherapy, and radiation therapy. A cohort of 100 women with breast cancer was included.

In 2020, in order to complete the findings of this quantitative interventional study, an ancillary qualitative study was conducted. This paper focuses on the findings of the later. Qualitative methods are indeed relevant in this context, they are a tool of choice for in-depth exploring how women with cancer experience these complementary therapies program. There is in fact a large qualitative literature analyzing the patients' lived experiences and views has explored the interest of complementary therapies as supportive care for women with breast cancer [8, 20, 21]. The qualitative literature exploring physical activity (structured exercise programs, pain-focused exercises, training sessions for weight management) has looked principally at the experience of breast cancer survivors and showed: 1) motivational factors influencing breast cancer survivors [22]; 2) the barriers and facilitators to patient participation in these interventions [23, 24]; and 3) their attitudes and beliefs regarding exercise programs [25, 26].

We found only one qualitative study exploring MBSR experience with a strict qualitative design [27], that is a phenomenological exploration of the lived experience of 8 women with stage I or II breast cancer who used MBSR. Four themes were identified: 1) the cancer journey: a shift in perception, 2) the treatment journey: the experience of MBSR, 3) the journey toward recovery, and 4) the journey toward self.

To our knowledge, there was no qualitative study exploring MBSR and physical activity among women with breast cancer within the French context.

The objective of this study was to explore the lived experience of women with breast cancer concerning this program of complementary therapies combining structured physical activity and the MBSR program.

## Materials and methods

This study followed the Inductive Process to analyze the Structure of lived Experience the (IPSE) [28], approach, a five-stage qualitative method tailored for clinical medical research. IPSE fits into the constructivist paradigm and is informed by a phenomenological approach. This approach is based on an inductive process designed to gain the closest access possible to the patients' experience, and to produce concrete recommendations.

The report of this study complies with the COREQ guideline [29]. This study was conducted from February to April 2021. This qualitative study is an ancillary study of the research

project "Effect of a personalized physical training program combined with a mindfulness-based mental training (MBSR) program on physical abilities and quality of life in patients after chemotherapy for breast cancer» **approved by the CPP (comité de protection des personnes/ committee for the protection of persons) EAST IV**—PRI 2014 HUS N˚5970, N˚ IDRCB: 2014-A01681-46. All participants provided informed written consent before inclusion.

## Stage 1: Setting up a research group

Our research group included three methodology experts, one man (JS), two women (ARL, EM), two women psychologists (JM, M-MV), three medical doctors (one woman oncologist, LV, and two general practitioners, one woman, VF, and one man, JPM) all experienced in qualitative research methods and two MBSR program instructors and doctors (EL, J-GB).

For heuristic purposes, the group's members were highly diverse, especially in their knowledge, age, and backgrounds. The group worked continuously on reflexivity during open discussions between the researchers.

## Stage 2: Ensuring the originality of the study

Two members of the group (ARL and VF) reviewed the qualitative and quantitative literature systematically and contacted other research teams that could most likely work on similar projects, to confirm the relevance and originality of the study. They verified that no qualitative study had ever been or was currently being conducted on exploring the lived experience of a similar program combining MBSR and physical activity. To remain inductive and open to novelty, the other group members had access to this review only after the data analysis had been completed.

## Stage 3: Recruitment and sampling, aiming for exemplarity

The research group defined the inclusion and exclusion criteria (**Table 1**). Sampling strategy was both purposive with maximum variation and convenient [30]:

- Purposively intended to attain exemplarity, that is, to select participants who have experienced archetypal examples of the situation being studied;

- Maximum variation of sample consisted of selection participants who differed by sex, age, family status, cancer stage, years of experience in other complementary therapies. That enabled the inclusion of participants who might enrich and add something new to what had previously been found;

The sampling strategy was also convenient, with a recruitment from the cohort of 100 women with breast cancer included in the prospective interventional study about the same program, facilitating the identification of breast cancer patients who had benefited from it.

**Table 1. Inclusion and exclusion criteria.**

| Inclusion criteria | Exclusion criteria |
|---|---|
| • Age: 18 years or older (no upper limit)<br>• had a breast cancer whatever the stage<br>• Cancer treatment completed<br>• Received supportive cancer care: a program of complementary therapies combining structured physical activity and MBSR<br>• Able to communicate in French | • Age: < 18 years |

The research group met regularly—usually after every 3 or 4 interviews—during the recruitment phase to select each new potential participant according to this purposive and convenient sampling strategy. After being selected by the group, EL (the principal investigator of the interventional study) contacted and recruited each participant directly. The interviews took place on average 5 years (a maximum of 9 and a minimum of 3 years) after the program.

Sample size was not defined in advance but was instead determined by data saturation according to the principles of "information power" [31]—here especially the criteria of the quality of dialogue during the interview and the sample specificity—and "theoretical sufficiency" [32], that is, data collection and analysis were complete when the group of researchers considered that the axes of experience obtained provided a sufficient explanatory framework for the data collected.

### Stage 4: Data collection, access to experience

Four researchers (JS, EM, JM, and M-MV) conducted in-depth face-to-face interviews with each participant after obtaining her consent and collecting social/demographic data. Before the study, we conducted four pilot interviews to determine our interview strategy. Because they revealed the fluidity of the participants' narratives as well as their apparent ease in relating their experience, we chose an open-ended approach. These women and their interviews were therefore included in the sample and the interviews structured around a single open question: *"Can you tell me the history of your disease and the treatments you've received from its start through today?"*

This approach suited our objective of obtaining an in-depth narrative of their lived experience; as the participants recalled their experience, the interviewers frequently prompted them to expand on their feelings, emotions, and thoughts. The interviewers used an interactive conversational style. The interviews lasted from 60 to 90 minutes. They were recorded and transcribed into anonymized verbatim, including the participants' expressive nuances. These transcripts were then analyzed.

### Stage 5: Data analysis, from the description of the structure of experience to practical implications

The IPSE analytic process is a rigorous procedure that relies on an inductive, phenomenological method [28]. In practice, the analysis had two stages: a stage of independent work by the three researchers and one of pooling the data collectively, by the group. The individual procedure consisted in three qualitative researchers (JS, EM, JM) independently and simultaneously conducting a systematic descriptive analysis aimed at conveying each participant's experience. This involved for each interview: 1) listening to the recorded interview twice and to reading it three times; 2) exploring the experience word by word, that is cutting up the entire text into descriptive units; 3) regrouping the descriptive units into categories. These stages are carried out with the help of QSR NVivo 12 software. During the group process, the three researchers met with the other members–familiarized with the data through listening and reading all the interviews—six times, in average after the analysis of five interviews, for two-hours meetings. The first group meetings were intended to conduct the structuring phase, that is, to regroup the categories into axes of experience, constructed such that each could be linked to its subjacent categories, and then to determine the structure of lived experience characterized by the central axes. During this structuring phase, the two members who reviewed the literature only intervened to discuss the originality and relevance–or the triviality- of each axis according to the literature. Then, the second set of meetings covered the practical phase, the process of

triangulation with the data in the literature that made it possible to identify the original aspects of the results and to suggest potential practical, clinical or research, implications.

### Criteria for rigor in the analyses and patient and public involvement

We used several criteria to ensure the rigor of the analysis:

- Data source triangulation, that is here the use of multiple data sources as a rigorous procedure to ensure a global understanding of the phenomenon under study.

- Investigator triangulation, with several researchers involved with data collection and individual analytical procedures.

- Attention to negative cases: Particular attention the cases in which new elements can differ radically from the emerging structure of the experience, and integration of these negative- sometimes contradictory- cases into the results.

- Reflexivity within the group process: the researchers' reflection of their role in the study and its effects on their findings at every step of the research process. This reflexive position is worked on constantly in the group, during open discussions between the researchers

- Feedback from "subjects of the experience" by presenting our results to a focus group of 8 patients, women who had been approached at the *Ligue contre le cancer* but not included in the sample. They all recognized their own experience in the structure we proposed.

## Results

The study included 29 women; their mean age was 52 years, with a range from 41 to 65 years. Participants' general characteristics are presented in **Table 2**, and detailed characteristics for each participant are available as (**S1 Table**). All participants easily accepted open discussion about their breast cancer, their treatment, and the supportive care program they received. The data analysis produced a structure of experience based on two central axes of these women's experience: 1) the experience they hoped for; and 2) an experience of discovery.

### 1. The hoped-for experience

Most participants described two principal expectations about supportive cancer care—that it would help them to: 1) take care of their bodies and themselves, 2) and become actors in their own care.

 **Take care of their bodies and themselves.** The participants described with much detail the effects and damage from their treatments, especially chemotherapy. Many felt "betrayed" by their bodies–P11: "that escapes and betrays me"- and, even before knowing about the complementary therapies program, looked for practical solutions that could "could help my body" and "act on the body "

> P13: *"At the beginning, I couldn't even go shopping. Finally, I couldn't do anything, nothing at all! [The standard treatments] destroys you completely, but completely! Physically, psychologically.*"

> P28: *"to put my body back in motion, to say, let's start over, get back on track.*"

 They all expressed the desire to restore their bodies, damaged and exhausted by both the disease and its treatments, and they were expecting that the program would have a direct

**Table 2. Summary of participants' characteristics.**

| Characteristics | N |
|---|---|
| **Gender** | |
| Women | 29 |
| **Age, mean** (years) | 52 y |
| **Year of diagnosis** | between 2013–2018 |
| **Treatments received** | |
| Surgery | 29 |
| Chemotherapy | 29 |
| Radiation therapy | 24 |
| Hormone therapy | 18 |
| **Complementary Therapies outside the supportive care** | |
| MBSR | 23 |
| Dietetics | 1 |
| Reconstructive/plastic surgical care | 4 |
| Sports | 25 |
| (*bicycle, rowing, Nordic walking, running, gym, tennis*) | |
| Mind/body activities | 10 |
| (*Pilates, yoga, tai chi, qi gong, reiki, reflexology, kinesiology, massage*) | |
| Thermal cure | 2 |
| Homeopathy | 7 |
| Psychological therapy | 12 |
| Physical therapy | 9 |

impact on their bodies. Indeed, they very often described the overall program as having allowed them to restore their confidence in their own bodies, to put their bodies damaged by the treatment back in shape, to bring them back to life, and to document their progress.

> P19: "*Exercise also lets you feel fully alive. To see that you are capable of doing things. Well I would never have spontaneously put myself on a bike, even if I had one. . . to tell yourself that you are not as weak as that, but that you are able, that I'm capable of doing perhaps more than I would have imagined.*"

*They also highlighted the self-care dimension, excepting to find within this program* another way to care for themselves and take care of themselves. According to them, this self-care dimension could not stand alone and had to be integrated in "the whole package that takes care of you", including the healthcare professionals. Similarly, several participants went beyond a dichotomy between "cure and care" in the treatment. The program was not seen as supportive care but was intended as being another line of the treatment, these two complementary therapies being considered as a full part of the therapeutic process, simultaneously adapted to and comprehensive against the disease.

> P11: "*It's the whole package that takes care of you. . . .having great doctors. I had fantastic surgeons, the care I had, the support from the program the whole time. It was great!*"

> P09: "*Sports and art saved me.*"

They also hoped that these complementary therapies, associated with conventional treatments, would reduce their side effects or would enable a better management, understanding

and acceptation of such effects (e.g., sensitivity to odors, fatigue, pain). They were hoping for "*perspective*" *(P17)*, psychological support, or even a diminution of their "angst".

> P10: *"And psychologically, I'd needed it. I think that if I had not had this support with exercise, psychologically I think I would have been worse, I would have stood it less well. Because I was coming out of that, I was content, I was happy, glad to be alive. It really helped me psychologically."*

Another strong idea emerged: the hope of accepting oneself with kindness. The kindness was conveyed by both activities—stationary biking and MBSR—though the idea of "absence of judgment." A majority felt its effects in their daily lives, years after the program, in their relationships with themselves.

> P23: *"Don't judge yourself anymore, be kind to yourself; this was very important."*

**1.2 Become actors in their own care, be proactive.** These women wanted to become "actors of their own care "by moving forward, volunteering for a study, and speaking up for the complementary therapies they received.

*First, the participants explained that they did not want to remain passive and maintain a situation of only enduring what was happening to them. In fact, most of the women reported taking an active position since the breaking of the diagnosis of breast cancer. They were willing to* "*move forward and not just endure it" (P15)*, "*to fight the disease"(P22)* and "*boost myself up" (P05). In order to do so, many participants "spontaneously and obviously" (P12) considered the use of complementary therapies.* Besides MBSR and physical activities, they also mentioned looking actively for dietetics, sophrology, homeopathy, tai-chi, or qi-qong. when they didn't find information at the hospital, these women turned toward local associations.

> P16: *"You had lectures on breast cancer, nutrition, sophrology, lots of things like that, sure. . . You could choose. Frankly, I advise lots of women to do this for gentle support of this whole treatment, which is actually very aggressive."*

> P12: *"And then I said: "ah, yes, yes, I'd really like to try that, it'll give me a good boost back up."*

Similarly, most of the participants were spontaneous volunteers to the interventional study. No one had to asked them to participate. They saw the flyer in a waiting room, or heard someone talking about the study, and they directly contacted the principal investigator with the will to participate to find supportive care through complementary therapies, but also *"to feel useful" and to be involved in a program that could lead to a care improvement of women with breast cancer.*

> P29: *"To participate in a study might be beneficial, plus if it can help move things forward, so much the better."*

They also highlighted how dense was the schedule of the program. They used the analogy of "a real job" (P2). Yet, they did not complain about it, rather they've seen it as an investment and as a good motivation for waking up in the morning and leaving the house, that is also to leave the house and come to the hospital for something else than medical care and medical examinations.

*P02*: *"It was real work, anyway, the meditation. Because you had to do things each week, you had to note everything, you had to write down the experiences you had, you had to. . . . anyway it's an investment, I mean, of yourself, of. . . that's what's interesting about it."*

*P08*: *"It was easier to wake up and drive until here, facing the morning traffic, knowing that it was to do sports and waiting hours for the chemo to end or for the doctor to show up".*

Finally, the participants, convinced by their own experience in the program, hoped that these complementary therapies would soon be integrated into the overall treatment proposal for breast cancer. They passed the information on to "reticent" physicians, other care providers, and patients. Participation in our interviews gave them the opportunity to pass their message on even further.

*P04*: *"I made a little book for the medical team to say*: *"look what I found, here is what you offer, and this is what you could offer."*

## 2. An experience of discovery

All the participants had had the experience of making important discoveries: 1) about themselves and 2) in their relationships with the exterior: that is, with others, society, and medical care.

**2.1 Discoveries about themselves.** In this context of disease and of treatments weakening them, making them more fragile, and causing them to lose physical capacities, the women interviewed described making surprising discoveries about their unsuspected physical and psychological capacities.

*P04*: *"After the first chemo sessions, the muscle. . . you have the impression that it's melting, it's. . . . and it happens fast. You lose your cardiac capacity and breathing and everything, very fast. And it's true that sometimes, I said*: *'for goodness sake, I climb the stairs at home, I feel like I'm climbing Mount Everest!"*

They underlined the experience of exertion, of "physical pain," and of combat associated both the physical activity (stationary biking) and the meditation. Yet, they reported (re)discovering the pleasure of physical exertion and regaining confidence in their bodies and in themselves. They felt alive, with bodies that "functioned" and were capable of progress. Some even talked about an "astonishing evolution" (P22).

*P12*: *"Being supported like that by doing sports, I think that can help anyway, because you want to fight. And stationary biking too, you want to. . . get better at it."*

*P25*: *"After, I had really great respiratory capacity, that felt good. You say to yourself: 'I'm not just a sick wreck, with only things wrong; my body still has parts that work!' and it's awesome!"*

Some experienced this program as tool for convalescence, to recover and to regain their life after the intensive treatment they endured. Other highlighted the "sublimating effects" of standard treatments, the complementary therapies of this program being described as a "catalyst" of the other treatments.

*P25*: *"after what I've been true with all of this strong treatment, what I need to get my life back was time for me, for my mind and my body (. . .) and this is what I got here."*

*P04: "Use of this medication was enriched by everything that resulted from the adapted exercise, the ability to meditate, it was like a sublimating effect of the basic medication use."*

Finally, they drew from this experience an undeniable—sometimes surprising—moral support in this trying period of cancer and its treatment.

*P26: "That gave me the opportunity to start again."*

*P10: "I almost forgot the disease. . .I didn't have any moment when nothing was going well; I was driving; it was unhoped for, I didn't think I would live the disease like that at all."*

One participant nonetheless described a persistent fatigue as an after-effect of the treatments, including the complementary therapies program, with important consequences on her return to work.

*P21 "The worst memory was really my return to work. I had lots of trouble concentrating, being tired. Afterward I was sick, a lot, because I was tired from what I had had before, because tired all the time is bad. . ."*

Not only the participants discovered or rediscovered physical and psychological capacities, they also reported having uncovered another way to embody their lives as women on a daily basis. They described the health care pathway as an occasion to revisit their bodies and to assert their places as women in the family and public space by having greater confidence in themselves.

*P02: "But meditation, it's like going back into yourself and saying 'I'm taking everything I've been through, I'm putting it inside me' and I'm saying to myself that I can trust myself."*

According to them, the program of complementary therapies played a particular role in this process. Some women allowed themselves to think of themselves in priority and others to detach themselves from the codes of femininity imposed by society and to affirm their choices, such as having short hair, being "rid of their bra," and feeling better, even liking themselves more.

P11: *"Now I'm beginning to think about myself. I tell myself: 'there's no reason that it's only for the others; I also have the right to think about me.' And before, I felt bad when I did that."*

*P13: "I don't wear a bra anymore. In fact, I've gotten rid of all my bras."*

Some also mentioned in addition other supportive care they experienced, especially in the associations (patient groups), and that played an important role in helping them "to live for [themselves]" and thus to reinforce their confidence in themselves and like themselves more. Other spoke about this program and other complementary therapies as a learning experience that transformed their lives and their outlook on life. They described lasting effects on their personal and professional lives and their relationships.

*P14: "I had a little more confidence in myself. . .I liked my body a little more."*

*P01: "And that's where you say to yourself: to take care of others, first you have to take care of yourself."*

Moreover, most of the participants reported that had "learned to listen to themselves more", to be more self-aware. They explained they were able to say "no" and to protect themselves. They would no longer forget themselves in their relationships.

P07: *"Because if I wear myself out doing the housework, I can't go walk for an hour. So I need to make a choice. And my choice, oh, well, I need to think about myself. That's what I didn't used to think."*

Some especially valued the effect of meditation. They mentioned having experienced a profound transformation of their interests, their outlook on life, even of their whole selves, reaching "a certain wisdom" in the way they were living their life, being more mindful of the moment, with more simplicity and serenity.

P18: *"Meditation taught me to live fully and simply, more serenely."*

P06: *"I continue meditation, with an app too, from time to time. And then my life, it's completely changed anyway. . . . I concentrate on my well-being because well it's something I need today and as a result. . . I've quit my job, through an economic layoff, and now I'm in training . . . to become a foot reflexologist."*

**2.2 Discovery in the relationship with the exterior: A new relationship experience.** The participants reported new relationship experiences through or thanks to the supportive care program, with family, friends, and healthcare workers

First, this program had both led them to discover a new, less judgmental way to listen to others and given them a desire to listen more. Some shared what they had learned during these activities, whether it was meditation or the helping relationship experienced with the nurse supervising the stationary biking sessions.

P24: *"No judgment. . .that still helps me today . . .I pay much more attention."*

P08: *"For me, it was important to transmit: 'here, I've been there, I could do that,' to keep myself aware of what's possible."*

Second, the lived experience of suffering from a cancer and going through such intensive treatment made them question their place within the world. According to most of them, the program helped them to redefine their place, as women, wives, and mothers.

P20: *"When disease arrives in your life. . ... It makes a break. It forced me to rest, it stopped me in my tracks, and I realized that I hadn't stopped running."*

On the one hand, the effects of exercise and MBSR made them "able to do sports," entitling them to return rapidly to normal life and rejoin a gym or other sports activities. On the other hand, some had made long-term commitments, investing in associations working on issues of breast cancer and femininity.

P04: *"And for club sports, my general practitioner didn't want to give me a 'certificate of fitness' to resume sports. He told me: 'it's too early, you can't.' So I had to explain what I had done and the certificate that Dr. Y had made saying I was fit."*

*P03*: *"And it's true that there was also this relation to femininity, which was very difficult and I had asked for there to be talks on this at the League (against Cancer), because no one talked about it. No one talked about that. Or that our bodies, they're completely ravaged."*

Their discoveries about themselves and this increased confidence in their bodies and their minds, as well as their capacity to invest themselves in totally new activities allowed them to feel "equal" for the first time in their lives and to be able to speak up in society. This experience of reciprocity in a protected space had induced them to prolong this experience in their daily lives and gain the confidence to talk about themselves.

*P07*: *"The day of silence, it's the rarest for me because, well, I talk, but I'm not really a chatterbox either. So I was able to sit down at a table with the doctor, he was at my table with other people who I had the impression . . . that they were more important than I was. And as there was no need to talk, I felt really equal, that day. And the fact that I didn't talk, that was. . . yes, it was a day. . . for me it was one of the best."*

Finally, all the participants underlined the crucial importance of feeling supported by trained and kind caregivers along their health care pathways, in a personalized fashion, completely safe at the hospital.
They recalled with emotion the sessions of meditation and stationary biking, with a program tailored for them. They felt cared for and "guided" in a context that was always pleasant and where the ambiance was always good.

*P26*: *"They knew me and they took care of me."*

*P09*: *"We talked a lot, it was very convivial; frankly it was great. I loved coming, really!"*

Added to the benefit of personalized support, the feeling of safety induced by sessions monitored by a nurse in a hospital setting was highlighted by many participants. Moreover, they were reassured that the overall program had been proposed by doctors and by health-care providers trained in these complementary therapies. The possibility of having someone there very regularly to answer their questions was also reassuring.

*P26*: *"The exercise with the bike, I pushed my body while I was surrounded by people; if I had been all alone, I would have worried. Even I felt safe. If something happened, Isabelle was there and the whole department behind her. My mind was really free, and I had nothing else to do."*

*P27*: *"Being able to ask questions twice a week of someone who knows and has experience with sick people, to be in contact with someone."*

The caring humanity of the staff and their relational quality had therapeutic functions. Most of the participants described the nurse supervising them in the stationary biking activity as "fabulous." Moreover, the end of the program marked the end of exercise for some, because of the impossibility of organizing this alone at home.

*P22*: *"Isabelle has lots of energy, and everything."*

*P15* *"So, Isabelle. . . it was above and beyond, frankly, she did it very very well, really. Smiling a lot. . . telling jokes. . . yes, always very warm and welcoming."*

## Discussion

This is the first qualitative study exploring in France the experience of a complementary therapies program—combining MBSR and structured physical activity -among women with breast cancer.

Two central axes of experience emerge from our analysis: (1) the experience they hoped for, that is to care for themselves and their bodies, and to become actors of the therapeutic process; and (2) an experience of discovery, with a self-discovery of their body, their mind, their women embodied identity, and a discovery of the relationships both within their personal space and in their health care space.

These results are in line with data described in the literature from other Western Educated, Industrialized, Rich and Democratic (WEIRD) countries.

- All the participants emphasized the need to be informed of the existence of supportive cancer care in their health-care pathways, and for this information to come from the health-care professionals themselves. This issue is already addressed in the literature [33], directly associated with the need to improve supportive cancer care access and resources, but also the coordination between the different participants and the integration of this care in conventional oncologic treatment [34, 35].

- Women's needs to be "listened to," "looked after," "paid attention to," "taken care of," and "cared for"—these all refer directly to the concept and ethics of care, as introduced by Gilligan and pursued since then [36].

- The need to target specific forms of supportive cancer care: improvement of cancer-related fatigue in breast cancer patients who attended yoga or Tai Chi sessions [12], improvement of quality of life when they received mind-body education [8] or art therapy [37], a significant improvement in physiological and cognitive functions, fatigue, emotional wellbeing, anxiety, depression, stress, distress, and mindfulness for the patients using an MBSR program [12].

Not only our results seem to be fully transferable to other contexts, but, in fact, all the qualitative studies conducted in a WEIRD country about complementary therapies as supportive cancer care, found close or similar results. The strength of qualitative research is to be situated and to be able to grasp an experience within a specific context. Similarly, the relevance of qualitative research is also to address all the complex aspects of a phenomenon, and we must admit that our results, as well as the qualitative literature on the matter, are quite simple and even superficial. In other words, we should have produced transferable complex results, not drawing a pseudo-universalist superficial account of the experience of these women.

This lack of originality in our results raises methodological questions about the evaluation and validation of these therapies. Applying the IPSE criteria of rigor, especially reflexivity, the analysis of the structure of experience we produced during the group meetings practical phase allowed us to have room for a methodological and reflexive deliberation to question properly the redundancy of results: Why did we end up with the same results as every other qualitative study on this subject? What does research with cancer patients mean to researchers? Are there areas of experience that we -and other qualitative scholars- avoided exploring despite our best intentions? why qualitative designs—including ours-, tailored for in-depth exploration, failed to address the complexity of this experience?

Our research team already conducted several qualitative studies in oncology, around cancer treatments and the QoL of cancer patients [38–40] but not about complementary therapies. In those earlier studies, we collected substantial data about the psychopathologic aspects linked to the experience of cancer, in particular, fear of death. Yet, in our results, neither the

psychopathologic aspects—emotional distress, uncertainty about the future, fears of the cancer spreading/returning, feelings of sadness, feelings about death and dying, feeling down or depressed—nor the question of death and the anxieties associated with it were mentioned by these participants. Nonetheless the fear of death is common among patients with cancer [41]. While there is no argument to support that recourse to CT is associated with higher levels of psychopathology and distress, some studies have found that for some patients the use of CT fulfils an important psychological need [42, 43]. These emotions were not mentioned in our interviews. It may be relevant that among the 29 women interviewed, 12 had simultaneous psychological support. One participant mentioned a reduction in anxieties as an expectation of this program, but without any other details and without any supplementary exploration by the interviewer. Of the eight patients attending the meeting where subjects received feedback about the study results, only one noted astonishment at the absence of data in our results about the fear of death associated with this cancer. On the one hand, a qualitative study has shown that using distraction, avoidance, and fewer coping skills is associated with a greater fear of cancer recurrence [44]. On the other hand, in a study assessing the effects of a death education program on fear of death, anxiety, and depression among breast cancer patients [45], the possibility of freely raising all the anxieties and questions around death had a soothing effect.

Moreover, there is a substantial literature on the unmet psychological needs of patients with cancer, with reports of unmet supportive care needs of patients with rare cancers [46, 47], of men with breast cancer [48], among older adults with cancer [49], in nutritional care in African pediatrics units [50], among people living with advanced cancer [51], and among indigenous cancer patients across Australia [52].

But why then did neither the participants, nor the researchers approach and explore these psychopathologic aspects and these psychological needs? The participants all reported a position toward the disease, one that echoes the principles of positive psychology, a psychological current promoting happiness, hope, flourishing, and fulfillment, all found in our results as well as in the literature: being strong in the face of disease, fighting and surviving, being positive and optimistic [52, 53]. Similarly, some women described the positive changes induced by their cancer experiences—recalling the concept of post-traumatic growth, another concept developed by positive psychology [54, 55]. A systematic review published in 2013 identified 16 studies reporting 5 types of positive psychology interventions used among patients with breast cancer, including mindfulness-based approaches, all promoting enhanced quality of life, well-being, hope, benefit finding, and optimism [56]. Mindfulness, stimulated by meditation or other exercises, is here considered as in tune with the current of positive psychology in the sense that it invites the patient to develop a positive state of mind in the face of life events [57]. Physical activity is also based on the principles of positive psychology, through various psychological hypotheses to explain its beneficial effects on mental health [58, 59].

It led us to think that, despite our inductive exploratory process and the "bracketing" (that is, our effort to identify and set aside) of our preconceptions, our research position is part of—implicitly and mirroring our participants—this positive psychology school of thought.

There are numerous criticisms of this psychological current, often from a Foucauldian or social constructivist perspective: positive psychology discourse is accused of shaping new subjectivities that fit into the program of neoliberal governmentality [60], positive psychology might deploy mechanisms to devalue humanistic psychology or privilege particular modes of functioning supporting a neoliberal economic and political discourse [61]. In this broader neoliberal project of positive psychology, individuals are seen as agents, responsible and accountable for their own well-being [62]. In 2005, Sundarajan published a critique of positive psychology using the donut metaphor, that is, life, happiness, and well-being as shaped by

positive psychology discourse is a donut: there is something missing at the core [63]. In line with these criticisms, our reflexive process has led us to think that, caught up in the discourse and the theory of positive psychology, the women, like the researchers, were not able to approach other aspects of the human experience, in particular, the aspects inherent in psychological distress and in existential and death-related anxieties.

Complementary therapies as supportive cancer care fall under a holistic perspective of the human being and seeks to "improve the quality of life of patients by preventing or treating the symptoms of the disease and the side effects caused by treatment of the disease. Our results illustrate the risk of a reduction when applying only a positivist vision.

Moreover, the absence of data on psychopathologic aspects in our interviews raises the question of compatibility with other supportive approaches theorized in psycho-oncology. Some authors have even considered the practice of the mindfulness as an "anti-elaborative" process, inhibiting secondary elaborative processing of thoughts, feelings, and sensations by regulating attention to bring a quality of non-elaborative awareness to current experience; attention is directed back to the breath, thereby preventing further elaboration [64]. In this research, this process and effect of mindfulness—avoiding elaboration or rumination—might have blocked the participants and researchers and prevented their access to aspects inherent in psychological distress and in the existential and death-related anxieties in our interviews.

## Research perspectives

This reflexive and theoretical elaboration led us to concrete research perspectives. Further qualitative research should be aware of and anticipate these inherent obstacles by providing original designs, such as exploring the experience of patients who dropped out complementary therapies as supportive care in cancer, focusing the exploration on the existential and psycho-pathological dimensions among patients with cancer doing a similar program, and investigating the incompatibility of elaborative and non-elaborative supportive care in cancer.

## Study limitations

Some limitations must be taken into consideration. First, our study took place in France and caution is required in transposing our results to other places, especially non-Western countries, because cancer care and supportive cancer care depend strongly on the organization of the medical system as well as on the country's economy.

Second, our recruitment process did not allow us to include the patients who had refused or stopped the supportive care activities proposed. This might have limited our findings.

Third, the mean age of the participants was 52 years with only 6 women younger than 50 years among the 29 interviewed. It would perhaps be interesting to vary the ages of the participants included in future studies. Further qualitative studies should be conducted to explore the important role age plays in these issues.

Finally, the IPSE approach postulates that the production of knowledge relies on intersubjectivity as a strategy for accessing valid knowledge of human real [28]. The researchers not addressing complex and difficult issues during the interviews could be partially seen as the pitfalls of such postulate.

## Conclusion

The results of our study and their redundancy with the results of others have allowed us to detail the theoretical framework of the positive psychology underlying complementary therapies in supportive care for breast cancer.

It seems primordial to us to maintain a diversity of underlying theoretical psychological approaches and not to reduce the complementary therapies to only this positivist current, which risks offering only "donut" therapies—that is, with something missing and that would miss the point for people with cancer.

## Supporting information

**S1 Table. Participants' characteristics.**
(DOCX)

## Acknowledgments

We would like to thank all the patients for their participation in this study. We also want to thank Jo Ann Cahn for the translation into English.

## Author Contributions

**Conceptualization:** Jordan Sibeoni, Evelyne Lonsdorfer-Wolf, Jean-Gérard Bloch, Anne Revah-Levy, Laurence Verneuil.

**Formal analysis:** Jordan Sibeoni, Valérie Feka, Anne Revah-Levy, Laurence Verneuil.

**Investigation:** Jordan Sibeoni, Emilie Manolios, Jeanne Mathé, Valérie Feka, Marie-Madeleine Vinez, Anne Revah-Levy, Laurence Verneuil.

**Methodology:** Jordan Sibeoni, Emilie Manolios, Jeanne Mathé, Marie-Madeleine Vinez, Franck Baylé, Anne Revah-Levy, Laurence Verneuil.

**Project administration:** Jean-Pierre Meunier.

**Resources:** Jean-Pierre Meunier.

**Supervision:** Evelyne Lonsdorfer-Wolf, Jean-Gérard Bloch, Anne Revah-Levy.

**Validation:** Jordan Sibeoni, Evelyne Lonsdorfer-Wolf, Jean-Pierre Meunier, Anne Revah-Levy.

**Writing – original draft:** Jordan Sibeoni, Emilie Manolios, Jeanne Mathé, Laurence Verneuil.

**Writing – review & editing:** Jordan Sibeoni, Emilie Manolios, Jeanne Mathé, Valérie Feka, Evelyne Lonsdorfer-Wolf, Jean-Gérard Bloch, Franck Baylé, Jean-Pierre Meunier, Anne Revah-Levy, Laurence Verneuil.

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
