## [Decision Letter · Decision Letter 0]

24 Jan 2023

PONE-D-22-28016The experience of a program combining two complementary therapies for women with breast cancer: an IPSE qualitative studyPLOS ONE

Dear Dr. Sibeoni,

Thank you for submitting your manuscript to PLOS ONE. After careful consideration, we feel that it has merit but does not fully meet PLOS ONE’s publication criteria as it currently stands. Therefore, we invite you to submit a revised version of the manuscript that addresses the points raised during the review process.

We look forward to receiving your revised manuscript.

Kind regards,

Adetayo Olorunlana, Ph.D.

Academic Editor

PLOS ONE

Journal Requirements:

3. Your abstract cannot contain citations. Please only include citations in the body text of the manuscript, and ensure that they remain in ascending numerical order on first mention.

Reviewers' comments:

Reviewer's Responses to Questions

**Comments to the Author**

1. Is the manuscript technically sound, and do the data support the conclusions?

Reviewer #1: Yes

Reviewer #2: Partly

Reviewer #3: Yes

2. Has the statistical analysis been performed appropriately and rigorously? 

Reviewer #1: N/A

Reviewer #2: N/A

Reviewer #3: N/A

3. Have the authors made all data underlying the findings in their manuscript fully available?

Reviewer #1: Yes

Reviewer #2: No

Reviewer #3: Yes

4. Is the manuscript presented in an intelligible fashion and written in standard English?

Reviewer #1: Yes

Reviewer #2: Yes

Reviewer #3: Yes

5. Review Comments to the Author

Reviewer #1: PONE-D-22-280 The experience of a program combining two complementary therapies for women with breast cancer: an IPSE qualitative study

1. The study presents the results of original research.

Yes, this is original research.

2. Results reported have not been published elsewhere.

Yes

3. Experiments, statistics, and other analyses are performed to a high technical standard and are described in sufficient detail.

High technical standard and in sufficient detail. The introduction part in the method section is identical with four to five other articles (all by the same authors). The text can be re-worded.

The aim was to explore the lived experience of women with breast cancer who received supportive cancer care through a program of complementary therapies combining structured physical activity and the MBSR program. The introduction is about breast cancer, supportive cancer and then some sentences about complementary therapies. There are some studies about these issues and the authors state; Moreover, no study has explored the global lived experience of a supportive care program based on combined complementary therapies among women with breast cancer. Our qualitative study aimed to fill this gap. I am doubtful that the authors will be able to explore the global lived experiences when interviewing 29 women with breast cancer.

When setting up the research group, why is there no oncology specialist? There are medical doctors from other specialties and psychologists and even two MBSR program

instructors and doctors. There could be a risk for directing the analysis.

In step 2. The reference 19 is using qualitative data nested within an evaluative randomised controlled trial (RCT). Data is from a questionnaire, so no pure qualitative research.

I understand that the authors want to introduce the” new method” but still it could be written as other qualitative methods- regarding the research process. The information below step 2 could have been in the introduction (you should always know what other studies there are in the field)

Step 3. Recruitment process OK. Sampling criteria—

1) select participants who have experienced archetypal examples of the situation being studied; Yes qualitative

2) include participants who might enrich and add something new to what had previously been found; How do you know?

3)facilitate the identification of breast cancer patients who had benefited from the programs, ok so you are looking for an evaluation of the program and not the lived experiences of breast cancer and complementary therapies?

4)be able to select participants who differed by sex, age, family status, years of experience, rank in their department, and type of practice.??? OK strategic samplings, but rank in department?? there is a need of clarification. There were 100 women in the “population” and 29 participated on this study, give some more information about who invited, how selected, did only 29 women out of these 100 responds.

I appreciate the use of information power and its” calculation.”

Step 4. Data collection in-depth interviews, one open question 60 to 90 minutes long.

Step 5. The method has quite a lot of influences, but it seems like the main idea of analysis is inductive description. Three researchers performed the analysis and then during the group

process, the three researchers met with the other members. Were the research ARL and VF participating? They did the literature review and could influence the result.

Trustworthiness’ is a mix from several methods and perspectives. A negative case was not presented, triangulation was only by the literature, not methodological or researcher triangulation. Peer review- subject experiences was performed, since it was descriptive and triangulated with the literature it had conformity and was easy to recognize.

The result- is two categories/axes only labelled similar to content analysis and then there is 2 sub-categories. Descriptive presentation with many quotations. But why sub-sub labels -- Restore their bodies, Take care of oneself, moving forward, volunteering for a

study, and speaking up for the complementary therapies they received. Discovery of their physical and psychological capacities, Another way of embodying her life as a woman on a daily basis, Learning to listen to oneself in a new and different way, In the environment of family and friends, In society. All these sub-sub categories send signals that data are not enough analysed. Sometimes there is a sub-subcategory with just one sentence and one quotation. Often there are sparse of information/data, so perhaps there should be more fluid text presenting the categories/axes. The quotations inform us as readers that this is about family and friends and so on.

Readers not used to qualitative research should benefit from a result clarified and presenting those categories/axes solid.

The discussion is repeating the result and confirmed by references used, nothing new presented, but this could be due to the research method- working with literature triangulation and systematic reviewing and then using focused research questions. –the lived experiences were not “identified” in the interviews.

The authors have also several systematic reviews about the research area- so the research questions are already reviewed.

It is appreciated that the authors have some reflections about these issues.

Delete the sentence about saturation it is NA and misplaced here.

Limitations are well presented but are lacking methodology issues, even though this method is new and rigorous, could there be some weakness?

4. Conclusions are presented in an appropriate fashion and are supported by the data.

Conclusions are presented in appropriate fashion, but it seems like the aim was to evaluate especially the MBSR program. The discussion and the conclusion end in a kind of theoretical paper focusing on theoretical psychological approaches.

5. The article is presented in an intelligible fashion and is written in standard English.

Yes, the article is presented in an intelligible fashion, but the structure could be sharpened, and it is written in standard English, mostly.

6. The research meets all applicable standards for the ethics of experimentation and research integrity.

Yes, this study meets all the applicable standards for research integrity.

7. The article adheres to appropriate reporting guidelines and community standards for data availability.

Yes, the article is following the reporting guidelines.

Out of 62 references 13 are 10 years old or more, there are 16 references in the introduction which all are up to date and out of these 6 are reviews. In total ten references are reviews, meta-analyses, or meta-synthesis and five references are a mix of methodology.

This could be an interesting paper, but it is hard to see what the focus is; a paper introducing this “new” method, evaluating complementary therapies and especially MBSR program or a theoretical discussion paper about psychological approaches. It is not obvious that this paper is about lived experiences…. This paper needs to be clarified, to be suitable for publication.

Reviewer #2: Thank you for your research. I have the follow suggestions to revise your manuscript.

Abstract

- Methods: The authors should add data collection date and inclusion and exclusion criteria. For example, did you include breast cancer patients with all stages? The authors should include how the data was analyzed.

- Results: It is hard to follow because there are two different types of numbers (e.g., 1, i). I recommend the authors consider the alphabet or verbally describe their findings.

Introduction

- Did the needed support differ by the cancer stage? Age? SES? Race/ethnicity?

- The authors should more literature review on previous studies that applied qualitative research methods. What were their main findings and advantages of using the methodology?

-Where is the study area (France? Any specific region)? The information should be indicated in the introduction section as well as the rationale for choosing that research area.

- The introduction was mainly about support, but your methodology focuses on physical activity. There needs to revise the manuscript more consistently.

Methods:

- The research question and the interview question are not aligned well.

- Although the authors mentioned, “The IPSE analytic process is detailed elsewhere”, I recommend they briefly describe the analytic process in this manuscript too.

- There should be a description of inter-rater reliability.

Findings:

- Instead of indicating P13, P28, the characteristics of these participants should appear in the main text with direct quotes (e.g., 53 years old woman with breast cancer stage 1).

- The findings are too shallow and descriptive. Instead of having multiple one-sentence quotes, the authors should have an in-depth analysis of the data.

- Wouldn’t there be any theoretical framework that the authors could apply?

Discussion:

- The implications should be expanded in the manuscript.

- Some sections that descriptively introduce the previous literature should be relocated to the introduction section.

Reviewer #3: This article is interesting, and the authors have referred to the COREQ.Due to the qualitative design, the authors did not perform statistical analysis, but they followed a certain qualitative analysis that may better be described in detail.  Authors also need to clarify on how to carry out the triangulation.

6. PLOS authors have the option to publish the peer review history of their article (what does this mean?). If published, this will include your full peer review and any attached files.

Reviewer #1: No

Reviewer #2: No

Reviewer #3: No

---

## [Author Response · Author response to Decision Letter 0]

6 Apr 2023

Point-by-point responses to reviewers

Reviewer #1

1. The introduction part in the method section is identical with four to five other articles (all by the same authors). The text can be re-worded.

Response: We modified the text accordingly.

Changes in the manuscript: page 6; This study followed the Inductive Process to analyze the Structure of lived Experience the (IPSE) [29], approach, a five-stage qualitative method tailored for clinical medical research. IPSE fits into the constructivist paradigm and is informed by a phenomenological approach. This approach is based on an inductive process designed to gain the closest access possible to the patients’ experience, and to produce concrete recommendations.

2. The aim was to explore the lived experience of women with breast cancer who received supportive cancer care through a program of complementary therapies combining structured physical activity and the MBSR program. The introduction is about breast cancer, supportive cancer and then some sentences about complementary therapies. There are some studies about these issues and the authors state; Moreover, no study has explored the global lived experience of a supportive care program based on combined complementary therapies among women with breast cancer. Our qualitative study aimed to fill this gap. I am doubtful that the authors will be able to explore the global lived experiences when interviewing 29 women with breast cancer.

Response: The reviewer is right, we should have better introduced the context of our study and focused on the program related with this research. We rewrote the introduction to make it more about the qualitative evaluation. We also decided to remove the term “global” that could bring some confusion. 

Changes in the manuscript: see the introduction section.

3. When setting up the research group, why is there no oncology specialist? There are medical doctors from other specialties and psychologists and even two MBSR program instructors and doctors. There could be a risk for directing the analysis.

Response: Among the three medical doctors, one is oncologist and two are general practioners. We should have mentioned their specialty. The heterogeneity of the group members, in terms of culture, knowledge, sex, age, occupation, and background aims to enrich the research at every stage, especially for the data analysis, so that the results are more robust and relevant and not limited to a single perspective.

Changes in the manuscript: page 6; Our research group included three methodology experts, one man (JS), two women (ARL, EM), two women psychologists (JM, M-MV), three medical doctors (one woman oncologist , LV, and two general practitioners, one woman ,VF, and one man, JPM) all experienced in qualitative research methods and two MBSR program instructors and doctors (EL, J-GB).

4. In step 2. The reference 19 is using qualitative data nested within an evaluative randomised controlled trial (RCT). Data is from a questionnaire, so no pure qualitative research.

Response: the reviewer is right. We decided to keep this reference but, following the reviewer’s suggestions, we moved all the literature -quantitative and qualitative- in the introduction. 

Changes in the manuscript: See the introduction.

5. I understand that the authors want to introduce the” new method” but still it could be written as other qualitative methods- regarding the research process. The information below step 2 could have been in the introduction (you should always know what other studies there are in the field)

Response: we followed the reviewer’s suggestion

Changes in the manuscript: See the introduction.

6. Step 3. Recruitment process OK. Sampling criteria—

1) select participants who have experienced archetypal examples of the situation being studied; Yes qualitative

2) include participants who might enrich and add something new to what had previously been found; How do you know?

Response: we thank the reviewer for this comment. We used the principle of maximum variation, meaning that point 2 (include participants who might enrich and add something new to what had previously been found) and 4 (be able to select participants who differed by sex…) were in fact intertwined. To clarify this methodological point, we rewrote this paragraph with the operationalization of the three principles we used (purposive, maximum variation and convenient)

Changes in the manuscript: page 7, Sampling strategy was both purposive with maximum variation and convenient [31]:

- Purposively intended to attain exemplarity, that is, to select participants who have experienced archetypal examples of the situation being studied; 

- Maximum variation of sample consisted of selection participants who differed by sex, age, family status, cancer stage, years of experience in other complementary therapies. That enabled the inclusion of participants who might enrich and add something new to what had previously been found;

The sampling strategy was also convenient, with a recruitment from the cohort of 100 women with breast cancer included in the prospective interventional study about the same program, facilitating the identification of breast cancer patients who had benefited from it.

3) facilitate the identification of breast cancer patients who had benefited from the programs, ok so you are looking for an evaluation of the program and not the lived experiences of breast cancer and complementary therapies?

Response: We understand what the reviewer meant, our first research question and objective was indeed a qualitative exploration of the lived experience of this program, seen as a paradigmatic illustration of complementary therapies in breast cancer support treatment (since there was a combination of two different therapies). However, we agree with the reviewer that our study should be more situated and contextualized and therefore our research question should remain only on the qualitative exploration of this program by women with breast cancer

We changed the introduction accordingly. 

Changes in the manuscript: see the Introduction section

4) be able to select participants who differed by sex, age, family status, years of experience, rank in their department, and type of practice.??? OK strategic samplings, but rank in department?? there is a need of clarification. There were 100 women in the “population” and 29 participated on this study, give some more information about who invited, how selected, did only 29 women out of these 100 responds.

Response: We are sorry for these mistakes due to the translation from French to English. What we meant by “rank of department” was “the cancer stage/severity”, as for type of practice it is related with the use of other complementary therapies outside of the program (as described in table 2.

For the recruitment of the 29 participants, we chose each participant together during the research meetings (based on the criteria we mentioned and the previous knowledge of the principal investigator of the interventional study (EL) that was also part of our research group. After selecting a new potential participant, also based on the previous inclusions, EL contacted her, presented the study protocol and organized the inclusion.

Changes in the manuscript: page 7

- Maximum variation of sample consisted of selection participants who differed by sex, age, family status, cancer stage, years of experience in other complementary therapies. That enabled the inclusion of participants who might enrich and add something new to what had previously been found;

The sampling strategy was also convenient, with a recruitment from the cohort of 100 women with breast cancer included in the prospective interventional study about the same program, facilitating the identification of breast cancer patients who had benefited from it.

The research group met regularly - usually after every 3 or 4 interviews - during the recruitment phase to select each new potential participant according to these 3 criteria. After being selected by the group, EL (the principal investigator of the interventional study) contacted and recruited each participant directly

7. Step 5. The method has quite a lot of influences, but it seems like the main idea of analysis is inductive description. Three researchers performed the analysis and then during the group process, the three researchers met with the other members. Were the research ARL and VF participating? They did the literature review and could influence the result.

Response: We thank the reviewer for these specific questions about the IPSE approach. All the points are addressed in the methodological paper. About the influences of IPSE, the method fits into the constructivist paradigm and is informed by a phenomenological approach. Inductive process is indeed at the very center of the approach. All stages of IPSE are informed by a phenomenological descriptive approach, not only the analytical procedure. As mentioned in the methodological paper (see Sibeoni et al. A Specific Method for Qualitative Medical Research: The IPSE (Inductive Process to Analyze the Structure of Lived Experience) Approach. BMC Med Res Methodol. 2020 Aug 26;20(1):216. https://doi.org/10.1186/s12874-020-01099-4.): “IPSE relies on an inductive process: the procedure is exploratory, and no research hypotheses are formulated before starting; rather, they emerge from the material, through methods designed to penetrate as far as possible into the participants’ lived experience. Because the data are collected and analyzed simultaneously, the analysis can affect the collection of the data, directly from the material, that is, the narrative of the participants’ lived experience.”

As for the participation of the members who did the literature review, again as mentioned in the methodological paper “To remain inductive and open to novelty, as mentioned above, the other group members have access to this review only after the data analysis has been completed. The tragedy of modern knowledge is, as Morin stated, that “the exponential increase in knowledge and references … stands in the way of reflecting on knowledge”. It is therefore important that physicians share the minimum of necessary knowledge to inform the study without impeding it by the curse of knowledge. 

The role of these two members during the group analysis phase is also specified: “It is very important during this phase that the physicians who analyzed the literature consider and discuss the originality and relevance of each axis, or on the contrary, its previous mentions or triviality according to the literature. »

Based on the reviewer’s comment, we decided to add this information into the method section.

Changes in the manuscript:

- Page 6: This study followed the Inductive Process to analyze the Structure of lived Experience the (IPSE) [29], approach, a five-stage qualitative method tailored for clinical medical research. IPSE fits into the constructivist paradigm and is informed by a phenomenological approach. This approach is based on an inductive process designed to gain the closest access possible to the patients’ experience, and to produce concrete recommendations.

- Page 8: The IPSE analytic process is a rigorous procedure that relies on an inductive, phenomenological method [29]. 

- Page 9: The first group meetings were intended to conduct the structuring phase, that is, to regroup the categories into axes of experience, constructed such that each could be linked to its subjacent categories, and then to determine the structure of lived experience characterized by the central axes. During this structuring phase, the two members who reviewed the literature only intervened to discuss the originality and relevance – or the triviality- of each axis according to the literature. Then, the second set of meetings covered the practical phase, the process of triangulation with the data in the literature that made it possible to identify the original aspects of the results and to suggest potential practical, clinical or research, implications.

8. Trustworthiness’ is a mix from several methods and perspectives. A negative case was not presented, triangulation was only by the literature, not methodological or researcher triangulation. Peer review- subject experiences was performed, since it was descriptive and triangulated with the literature it had conformity and was easy to recognize.

Response: We thank the reviewer for this relevant comment. We removed the word trustworthiness to avoid methodological confusion. We also added the definition of our criteria and specify our triangulation process– that was indeed only with the literature- and attention to negative cases –that is paying attention to find new elements that differed radically from the emerging structure of experience. If a case differs completely from the proposed structure of the experience, the IPSE approach considers that theoretical sufficiency has not been reached and new interviews and analyses need to be conducted. As for the feedback to subjects of experience, we understand what the reviewer means: our findings are in line with the literature data and didn’t produce any effects of surprise when presenting during the focus group. Yet, this ensured the transferability of our results into the French context. 

Changes in the manuscript: page 9 

We used several criteria to ensure the rigor of the analysis:

- Data source triangulation, that is here the use of multiple data sources as a rigorous procedure to ensure a global understanding of the phenomenon under study.

- Investigator triangulation, with several researchers involved with data collection and individual analytical procedures.

- Attention to negative cases: Particular attention the cases in which new elements can differ radically from the emerging structure of the experience, and integration of these negative- sometimes contradictory- cases into the results. 

- Reflexivity within the group process: the researchers’ reflection of their role in the study and its effects on their findings at every step of the research process. This reflexive position is worked on constantly in the group, during open discussions between the researchers.

9. The result- is two categories/axes only labelled similar to content analysis and then there is 2 sub-categories. Descriptive presentation with many quotations. But why sub-sub labels -- Restore their bodies, Take care of oneself, moving forward, volunteering for a study, and speaking up for the complementary therapies they received. Discovery of their physical and psychological capacities, Another way of embodying her life as a woman on a daily basis, Learning to listen to oneself in a new and different way, In the environment of family and friends, In society. All these sub-sub categories send signals that data are not enough analysed. Sometimes there is a sub-subcategory with just one sentence and one quotation. Often there are sparse of information/data, so perhaps there should be more fluid text presenting the categories/axes. The quotations inform us as readers that this is about family and friends and so on.

Readers not used to qualitative research should benefit from a result clarified and presenting those categories/axes solid.

Response: The reviewer is right about the presentation of our results. We modified the presentation of the results accordingly, removing all the sub-sub categories and presenting the categories with more details in the shape of a fluid text. We understand that it could give a “false” signal of insufficient data analysis, however, we would like to highlight that in an IPSE study, the results are the presentation of the structure of experience and not an exhaustive presentation of the thematic analysis. In the methodological paper, the authors note that “Exhaustive results, unranked, may dilute the original points and the new information, thus impeding any translation of the results into direct implications”. 

Changes in the manuscript: See the results section.

10. The discussion is repeating the result and confirmed by references used, nothing new presented, but this could be due to the research method- working with literature triangulation and systematic reviewing and then using focused research questions. –the lived experiences were not “identified” in the interviews. The authors have also several systematic reviews about the research area- so the research questions are already reviewed.

Response: We thank the reviewer for this comment that gave the opportunity to explain further the IPSE approach that is meant to uncover original findings, using an inductive approach and structuring the triangulation process in a way that would induce cognitive bias such as “confirmation bias”, “selection bias” or “curse of knowledge”. We want to highlight that triangulation occurred during the last phase of the analytic process to avoid such bias and facilitate the emergence and production of original results. In front of such redundancy, we decided to in-depth investigate through individual and group reflexivity, asking ourselves relevant questions: Why did we end up with the same results as every other qualitative study on this subject? What does research with cancer patients mean to researchers? Are there areas of experience that we -and other qualitative scholars- avoided exploring despite our best intentions? why qualitative designs - including ours-, tailored for in-depth exploration, failed to address the complexity of this experience? 

We are not sure what the reviewer meant by “identification” of lived experiences. Our definition of lived experience is in tune with experiential qualitative approach, that is lived experience being defined as personal knowledge of the world gained through direct participation and involvement in the event or phenomenon. Lived experience refers to human activities that are immediate, situated and daily, which are lived without thinking about or paying attention to them (pre-reflexive experience). We agree, however, that a part of the experience is not shared, expressed and maybe even thought; and this is what we wanted to discuss. 

We agreed with the reviewer that this research question has been the object of qualitative explorations, yet not in the French context. We did not insist enough about the context since our findings were quite similar with results from qualitative studies conducted in English-spoken countries, so transferable. Building on the reviewer’s comment, we changed the discussion to introduce this contextual dimension and explain better how the criteria of rigor used in the IPSE approach helped us to address relevant issues regarding complementary therapies in supportive care of cancer. 

Changes in the manuscript: See the discussion section

11. It is appreciated that the authors have some reflections about these issues.

Delete the sentence about saturation it is NA and misplaced here.

Limitations are well presented but are lacking methodology issues, even though this method is new and rigorous, could there be some weakness?

Response: the reviewer is right, we also need to add as limitations, the one that are intrinsic to the IPSE method. This approach postulates that the production of knowledge relies on three elements: “(i) subjectivity as a space for constructing human reality, (ii) intersubjectivity as a strategy for accessing valid knowledge of human reality, and (iii) understanding that human reality takes place in daily life.“

We think that this research shows the potential pitfalls of the intersubjective approach with what can be understood as a neutralization during the interviews that impede the interviewers to address more complex and difficult issues during the interviews such as death, depression, and so on. 

We added this methodological issue in the limitation section.

Changes in the manuscript: page 25 

Finally, the IPSE approach postulates that the production of knowledge relies on intersubjectivity as a strategy for accessing valid knowledge of human real [29]. The researchers not addressing complex and difficult issues during the interviews could be partially seen as the pitfalls of such postulate.

4. Conclusions are presented in an appropriate fashion and are supported by the data.

Conclusions are presented in appropriate fashion, but it seems like the aim was to evaluate especially the MBSR program. The discussion and the conclusion end in a kind of theoretical paper focusing on theoretical psychological approaches.

Response: we already answered and modified accordingly the manuscript to redefine the scope of our study that was not as broad as mentioned.

We agree with the reviewer that our discussion has a theoretical part but we consider it as the rationale to reach concrete research perspectives. We added a section in the discussion with this sub-heading title “Research perspectives” to insist of the fact that the goal was not to provide some theoretical views but to find ways to improve the quality of qualitative research being done in this field.

Changes in the manuscript: page 24, 25

Research perspectives

This reflexive and theoretical elaboration led us to concrete research perspectives. Further qualitative research should be aware of and anticipate these inherent obstacles by providing original designs, such as exploring the experience of patients who dropped out complementary therapies as supportive care in cancer, focusing the exploration on the existential and psychopathological dimensions among patients with cancer doing a similar program, and

investigating the incompatibility of elaborative and non-elaborative supportive care in cancer

5. The article is presented in an intelligible fashion and is written in standard English.

Yes, the article is presented in an intelligible fashion, but the structure could be sharpened, and it is written in standard English, mostly.

Response: We hope that all the changes we made, integrating the reviewer’s comments and suggestions, have sharpened the structure of the paper.

Reviewer #2:

Abstract

- Methods: The authors should add data collection date and inclusion and exclusion criteria. For example, did you include breast cancer patients with all stages? The authors should include how the data was analyzed.

Results: It is hard to follow because there are two different types of numbers (e.g., 1, i). I recommend the authors consider the alphabet or verbally describe their findings.

Response: we followed the reviewer’s suggestion and made the changes accordingly.

Changes in the manuscript: page 2, abstract 

Abstract

Introduction: The use of complementary therapies within oncology is a clinical issue, and their evaluation a methodological challenge. This paper reports the findings of a qualitative study exploring the lived experience of a French program of complementary therapies combining structured physical activity and MBSR among women with breast cancer.

Methods: This French exploratory qualitative study followed the five stages of the Inductive Process to analyze the Structure of lived Experience (IPSE) approach. Data was collected from February to April 2021 through semi structured interviews. Participants, purposively selected until data saturation. Inclusion criteria were: being an adult woman with breast cancer whatever the stage who had completed their treatment and were part of the program of complementary therapies. 

Results: 29 participants were included. Data analysis produced a structure of experience based on two central axes : 1) the experience these women hoped for, with two principal expectations, that is to take care of their bodies and themselves, and to become actors in their own care; and 2) an experience of discovery, first of themselves and also in their relationship with the exterior, whether with others, or in society, and in the relationships with health-care providers.

Conclusions: Our results from this French study reinforce the data described in other western countries about the needs of women receiving care in oncology departments for breast cancer: they need to be informed of the existence of supportive care in cancer by the health-care professionals themselves, to be listened to, and to receive support care. A systematic work of reflexivity about this redundancy in our results and in the qualitative literature, led us to question what impeded the exploration of more complex aspects of the experience of this women - the inherently emotional and anxiety-inducing experience of cancer, especially anxiety about its recurrence and of death – and to suggest new research perspectives to overcome these methodological and theoretical obstacles.

Introduction

- Did the needed support differ by the cancer stage? Age? SES? Race/ethnicity?

Response: we thank the reviewer for this question. We added in the introduction the information we found and the references. We would like also to indicate to the reviewer that in France, data about race and ethnicity are not allowed according to the French principles of laicity.

Changes in the manuscript: Page 4 

Clinical practice guidelines from the Society for Integrative Oncology on the use of integrative therapies during and after breast cancer treatment recommend especially mind-body therapies but do not give any clinical indications or factors (age, stage of cancer, socioeconomic status) to choose among the many supportive care strategies [15]. However, one qualitative study conducted in United States has shown the influence of socio-ecological and cultural factors (beliefs about the illness, gender roles and family obligations) on the health-related quality of life of women with breast cancer [16].

- The authors should do more literature review on previous studies that applied qualitative research methods. What were their main findings and advantages of using the methodology?

Response: the literature review was in the method section (Step2) but, following the reviewer’s advice, we moved it in the introduction. 

Changes in the manuscript: see the introduction section

-Where is the study area (France? Any specific region)? The information should be indicated in the introduction section as well as the rationale for choosing that research area.

Response: We added this information in the introduction

Changes in the manuscript: Page 4; In 2015, women with breast cancer being treated in oncology departments of the university hospital of Strasbourg, situated at the border with Germany…

- The introduction was mainly about support, but your methodology focuses on physical activity. There needs to revise the manuscript more consistently.

Response: The reviewer is right. Our initial research question is directly related with the program combining MBSR and physical activity. Therefore, we made some substantial changes in the introduction to better contextualize our qualitative study.

Changes in the manuscript: See Introduction section

Methods:

- The research question and the interview question are not aligned well.

Response: We thank the reviewer for this comment that gives us the opportunity to explain further an important point concerning the IPSE method. The comment of the reviewer would be relevant if we had used a qualitative elicitation research approach, based on directive, task-oriented interviews. The IPSE approach doesn’t restrict the direction of the conversations for both participants and interviewers. It instead keep the research open to what the participants’ narratives of the experience can add, to allow them to share what they have lived. The goal of the data collection is to reach the narrative of the experience, the tool used to obtain this narrative, for instance here an open question about their illness history, always depends on the context. This first question is to be considered as a tool, a narrative support to get access later during the interview on spontaneous narratives of experiences regarding the phenomenon under study. 

- Although the authors mentioned, “The IPSE analytic process is detailed elsewhere”, I recommend they briefly describe the analytic process in this manuscript too.

Response: We followed the reviewer idea and added more details regarding the analytic process

Changes in the manuscript: page 8, 9

The IPSE analytic process is a rigorous procedure that relies on an inductive, phenomenological method [29]. In practice, the analysis had two stages: a stage of independent work by the three researchers and one of pooling the data collectively, by the group. The individual procedure consisted in three qualitative researchers (JS, EM, JM) independently and simultaneously conducting a systematic descriptive analysis aimed at conveying each participant’s experience. This involved for each interview: 1) listening to the recorded interview twice and to reading it three times; 2) exploring the experience word by word, that is cutting up the entire text into descriptive units; 3) regrouping the descriptive units into categories. These stages are carried out with the help of QSR NVivo 12 software. During the group process, the three researchers met with the other members –familiarized with the data through listening and reading all the interviews - six times, in average after the analysis of five interviews, for two-hours meetings. The first group meetings were intended to conduct the structuring phase, that is, to regroup the categories into axes of experience, constructed such that each could be linked to its subjacent categories, and then to determine the structure of lived experience characterized by the central axes. During this structuring phase, the two members who reviewed the literature only intervened to discuss the originality and relevance – or the triviality- of each axis according to the literature. Then, the second set of meetings covered the practical phase, the process of triangulation with the data in the literature that made it possible to identify the original aspects of the results and to suggest potential practical, clinical or research, implications.

- There should be a description of inter-rater reliability.

Response: inter-rater reliability cannot be used in an IPSE qualitative study, the analytic procedure being inductive with no pre-established codes. However, we fully described the group process and the criteria of rigor of this approach.

Findings:

- Instead of indicating P13, P28, the characteristics of these participants should appear in the main text with direct quotes (e.g., 53 years old woman with breast cancer stage 1).

Response: We understand why the reviewer asked for this information in the main text. However we are afraid that it would make the results less readable and we want to aslo privilege the comfort of the readers.

- The findings are too shallow and descriptive. Instead of having multiple one-sentence quotes, the authors should have an in-depth analysis of the data.

Response: Our approach is descriptive and not interpretative, so the findings need to remain descriptive. They are in fact quite superficial, and this is a point we added in the discussion thanks to the reviewer. We also rewrote the result section to give more findings from our in-depth analysis and avoid this impression of multiple one-sentence quotes. 

Changes in the manuscript: 

See the results section.

page 21, Not only our results seem to be fully transferable to other contexts, but, in fact, all the qualitative studies conducted in a WEIRD country about complementary therapies as supportive cancer care, found close or similar results. The strength of qualitative research is to be situated and to be able to grasp an experience within a specific context. Similarly, the relevance of qualitative research is also to address all the complex aspects of a phenomenon, and we must admit that our results, as well as the qualitative literature on the matter, are quite simple and even superficial. In other words, we should have produced transferable complex results, not drawing a pseudo-universalist superficial account of the experience of these women. 

- Wouldn’t there be any theoretical framework that the authors could apply?

Response: Yes, the IPSE approach fits into the constructivist paradigm and is informed by a phenomenological approach. All stages of IPSE are informed by a phenomenological descriptive approach, (see Sibeoni et al. A Specific Method for Qualitative Medical Research: The IPSE (Inductive Process to Analyze the Structure of Lived Experience) Approach. BMC Med Res Methodol. 2020 Aug 26;20(1):216. https://doi.org/10.1186/s12874-020-01099-4.

Changes in the manuscript: Page 6

This study followed the Inductive Process to analyze the Structure of lived Experience the (IPSE) [29], approach, a five-stage qualitative method tailored for clinical medical research. IPSE fits into the constructivist paradigm and is informed by a phenomenological approach

Discussion:

- The implications should be expanded in the manuscript.

Response: We follow the reviewer’s suggestion and added an “research perspective” section in the discussion with the concrete implications drawn from our study

Changes in the manuscript: Page 24, 

Research perspectives

This reflexive and theoretical elaboration led us to concrete research perspectives. Further qualitative research should be aware of and anticipate these inherent obstacles by providing original designs, such as exploring the experience of patients who dropped out complementary therapies as supportive care in cancer, focusing the exploration on the existential and psychopathological dimensions among patients with cancer doing a similar program, and

investigating the incompatibility of elaborative and non-elaborative supportive care in cancer. 

- Some sections that descriptively introduce the previous literature should be relocated to the introduction section.

Response: done

Reviewer #3

This article is interesting, and the authors have referred to the COREQ. Due to the qualitative design, the authors did not perform statistical analysis, but they followed a certain qualitative analysis that may better be described in detail. 

Response: We thank the reviewer for this suggestion. We described with more details the analytic procedure

Changes in the manuscript: page 8, 9

The IPSE analytic process is a rigorous procedure that relies on an inductive, phenomenological method [29]. In practice, the analysis had two stages: a stage of independent work by the three researchers and one of pooling the data collectively, by the group. The individual procedure consisted in three qualitative researchers (JS, EM, JM) independently and simultaneously conducting a systematic descriptive analysis aimed at conveying each participant’s experience. This involved for each interview: 1) listening to the recorded interview twice and to reading it three times; 2) exploring the experience word by word, that is cutting up the entire text into descriptive units; 3) regrouping the descriptive units into categories. These stages are carried out with the help of QSR NVivo 12 software. During the group process, the three researchers met with the other members –familiarized with the data through listening and reading all the interviews - six times, in average after the analysis of five interviews, for two-hours meetings. The first group meetings were intended to conduct the structuring phase, that is, to regroup the categories into axes of experience, constructed such that each could be linked to its subjacent categories, and then to determine the structure of lived experience characterized by the central axes. During this structuring phase, the two members who reviewed the literature only intervened to discuss the originality and relevance – or the triviality- of each axis according to the literature. Then, the second set of meetings covered the practical phase, the process of triangulation with the data in the literature that made it possible to identify the original aspects of the results and to suggest potential practical, clinical or research, implications.

 Authors also need to clarify on how to carry out the triangulation.

Response: the reviewer is right. We defined better the two types of triangulation we used – investigator triangulation, and data source triangulation- and added sentences to clarify the role played by the literature (data source triangulation). 

Changes in the manuscript : Page 9 

During this structuring phase, the two members who reviewed the literature only intervened to discuss the originality and relevance – or the triviality- of each axis according to the literature. Then, the second set of meetings covered the practical phase, the process of triangulation with the data in the literature that made it possible to identify the original aspects of the results and to suggest potential practical, clinical or research, implications.

Data source triangulation, that is here the use of multiple data sources as a rigorous procedure to ensure a global understanding of the phenomenon under study.

- Investigator triangulation, with several researchers involved with data collection and individual analytical procedures.

---

## [Decision Letter · Decision Letter 1]

27 Apr 2023

The experience of a program combining two complementary therapies for women with breast cancer: an IPSE qualitative study

PONE-D-22-28016R1

Dear Dr. Sibeoni,

We’re pleased to inform you that your manuscript has been judged scientifically suitable for publication and will be formally accepted for publication once it meets all outstanding technical requirements.

Kind regards,

Adetayo Olorunlana, Ph.D.

Academic Editor

PLOS ONE

Additional Editor Comments (optional):

Reviewers' comments:

Reviewer's Responses to Questions

**Comments to the Author**

1. If the authors have adequately addressed your comments raised in a previous round of review and you feel that this manuscript is now acceptable for publication, you may indicate that here to bypass the “Comments to the Author” section, enter your conflict of interest statement in the “Confidential to Editor” section, and submit your "Accept" recommendation.

Reviewer #1: All comments have been addressed

Reviewer #2: All comments have been addressed

2. Is the manuscript technically sound, and do the data support the conclusions?

Reviewer #1: Yes

Reviewer #2: Yes

3. Has the statistical analysis been performed appropriately and rigorously? 

Reviewer #1: N/A

Reviewer #2: Yes

4. Have the authors made all data underlying the findings in their manuscript fully available?

Reviewer #1: Yes

Reviewer #2: Yes

5. Is the manuscript presented in an intelligible fashion and written in standard English?

Reviewer #1: Yes

Reviewer #2: Yes

6. Review Comments to the Author

Reviewer #1: Thank you for all efforts in amending the manuscript. Mostly comments from reviewers are taken into consideration.

Just a small thing; saturation is still in the abstract. This is not applicable to IPSE or Content analysis. Saturation demands parallel data collection and analysis. Perhaps you want to use the term redundancy.

Thank you for interesting reading.

Reviewer #2: THe authors revised the manuscript well, incorporating reviewers' comments. I recommend the manuscript for publication.

7. PLOS authors have the option to publish the peer review history of their article (what does this mean?). If published, this will include your full peer review and any attached files.

Reviewer #1: No

Reviewer #2: No

---

## [Editor Report · Acceptance letter]

9 May 2023

PONE-D-22-28016R1 

The experience of a program combining two complementary therapies for women with breast cancer: an IPSE qualitative study 

Dear Dr. Sibeoni:

I'm pleased to inform you that your manuscript has been deemed suitable for publication in PLOS ONE. Congratulations! Your manuscript is now with our production department. 

Kind regards, 

on behalf of

Associate Professor Adetayo Olorunlana 

Academic Editor

PLOS ONE